# Iterative rolling difference-Z-score and machine learning imputation for wind turbine foundation monitoring

Renjie Li[1], Xiangxing Lu[1], Jizhang Zhao[2,3,4]*, Weibing Chen[1], Huanwei Wei[2,3,4], Cong Liu[2,3,4]

1 Shandong Electric Power Engineering Consulting Institute Corp., Ltd., Jinan, China, 2 School of Civil Engineering, Shandong Jianzhu University, Jinan, China, 3 Key Laboratory of Building Structural Retrofitting and Underground Space Engineering, Ministry of Education, Jinan, China, 4 Subway Protection Research Institute, Shandong Jianzhu University, Jinan, China

* saturninasyrop6@gmail.com

## Abstract

In engineering structure performance monitoring, capturing real-time on-site data and conducting precise analysis are critical for assessing structural condition and safety. However, equipment instability and complex on-site environments often lead to data anomalies and gaps, hindering accurate performance evaluation. This study, conducted within a wind farm reinforcement project in Shandong Province, addresses these challenges by focusing on anomaly detection and data imputation for weld nail strain, anchor cable axial force, and concrete strain. We propose an innovative iterative rolling difference-Z-score method for anomaly detection and a machine learning-based imputation framework combining linear interpolation with LightGBM. Experimental results show that the iterative rolling difference-Z-score method detects single-point and clustered anomalies with a Z-score threshold of 4, achieving robust performance even with 80% data loss. The imputation framework maintains low mean squared error (MSE) of 0.0214–0.0227 and root mean squared error (RMSE) of 0.14–0.15 for continuous missing data scenarios (60–200 points), with reliable reconstruction up to 50% data loss. This research provides a robust solution for ensuring the precision and integrity of wind farm monitoring data, enhancing long-term structural reliability in renewable energy applications.

## Introduction

In the increasingly competitive landscape of new energy technologies, wind power has become a significant global choice due to its clean and renewable nature [1]. The foundation of a wind turbine is critical to its operation, as it supports the turbine's weight and withstands dynamic loads during operation [2–4]. The strength and stability of the turbine foundation directly affect the turbine's functionality. Given the

**Data availability statement:** All relevant data are within the paper and its Supporting Information files.

**Funding:** The author(s) received no specific funding for this work.

**Competing interests:** The authors have declared that no competing interests exist.

prevalence of turbine failures attributable to foundation damage, especially in older models, reinforcing the turbine foundation is crucial for ensuring long-term stable operation [5].

In recent years, substantial progress has been made in the maintenance and reinforcement of turbine foundations. For instance, Zhao et al. [6] conducted extensive field monitoring and numerical analysis of large wind turbine reinforced concrete foundations, revealing changes in reinforcement effects under environmental loads. Gondle et al. [7] developed a low-cost mechanical displacement indicator and corresponding finite element models to assess the long-term performance and potential degradation of turbine foundation systems. Wei et al. [8] used field monitoring methods to analyze the performance of rock anchor-based turbine foundation reinforcement.

During the long-term monitoring of wind turbine foundations, environmental factors may cause abnormal data fluctuations, and electromagnetic interference can compromise data transmission accuracy, leading to abnormal monitoring data. Additionally, unstable power supplies or sensor damage due to environmental pressures may result in data loss. Addressing these issues is a significant challenge in long-term monitoring [9].

Currently, standard methods for anomaly detection [10] include Isolation Forest, One-Class SVM, DBSCAN, Local Outlier Factor (LOF), K-Means, Gaussian Mixture Model, Autoencoder, and Random Projection. Machine learning-based anomaly monitoring methods use models such as Convolutional Neural Networks (CNN) [11], which transform detection problems into classification or prediction tasks, including models like Quantile Regression Neural Networks (QRNN) [12], Artificial Neural Networks (ANN) combined with error analysis [13], Stacked Adversarial Variational Recurrent Neural Networks (SAVRNN) [14], and combinations of Recurrent Neural Networks (RNN) with Long Short-Term Memory (LSTM) [15–17] for anomaly detection. Standard methods for handling missing data include statistical methods and machine learning techniques.

However, these algorithms often suffer from poor interpretability and limited transferability in practical engineering projects; while they can identify anomalies, they fail to explain the causes or probabilities of these anomalies, and training a neural network is difficult to transfer to other datasets. To address this issue, this paper proposes a new anomaly detection method that effectively solves anomaly detection problems in practical engineering projects. Statistical methods rely on the statistical characteristics of the dataset to infer and fill in missing values, suitable for simpler data distributions [18,19]. Common techniques in this domain include maximum expectation filling [20], regression filling [20], and multiple imputation [21]. In contrast, machine learning methods [22] take a more dynamic and complex approach, treating missing values as target attributes that need to be predicted and filled [23–29].

This study, based on a wind farm reinforcement project in Shandong Province, explores how to manage data anomalies and data loss encountered during long-term monitoring. For monitoring data anomalies, we propose an Iterative Rolling Difference-Z-score method for anomaly detection, which effectively addresses large-scale continuous data anomalies caused by sensor failures and extensive data loss. For monitoring data

loss, we introduce a new data imputation framework combining linear interpolation, machine learning, and the Iterative Rolling Difference-Z-score, providing a robust guarantee for the accuracy and integrity of wind farm monitoring data.

## Engineering example analysis

### Project overview

The wind farm consists of 24 wind turbine units, each supported by foundations constructed using secondary grouting micro-piles and base platforms. The wind farm began operations and grid-connected power generation in November 2013. During a maintenance check in early 2021, it was discovered that over 180 internal anchor rods and more than 20 external anchor rods in the turbines had fractured, with the numbers continuing to rise. Fig 1(a) illustrates the situation of an internal anchor rod fracture. Additionally, the turbine foundations show radial and circumferential cracks of varying depths and widths, with radial cracks being predominant and circumferential cracks concentrated in the root area, as shown in Fig 1(b).

Given the significant safety hazards posed by existing foundation issues affecting the operational integrity of the units, an informed decision was undertaken to initiate experimental reinforcement work on Turbine #1. The proposed reinforcement design for the turbine foundation encompasses the integration of weld nails into the tower structure and the application of an external concrete encasement at the base. Specifically, weld nails will be strategically positioned within a defined height range at the tower's base, accompanied by an external concrete layer encasing both the existing foundation and the weld-nailed section of the tower. The new and existing concrete foundations will be integrally connected via pressure-dispersive anchor rods extending into the ground. The construction site designated for the reinforcement is illustrated in Fig 2(a), with a detailed schematic of the reinforcement plan presented in Fig 2(b).

### Arrangement of monitoring points

To evaluate the structural safety and performance enhancement post-reinforcement of the turbines, it is essential to ascertain the force patterns and load distribution of the turbine foundation following the upgrade. Within this project, strain gauges and anchor assemblies were embedded during construction to facilitate comprehensive monitoring. The specific structural monitoring scope encompasses the external concrete, foundation reinforcing bars, external concrete sidewall rebars, newly added weld nail forces, and axial forces of the anchor cables. The arrangement of monitoring points is as follows: strain gauges for weld nail monitoring are strategically placed in the weld nail region on the outer surface of the wind turbine tower, with the planes designated as D1 to D5. In the non-door area, nine monitoring points are uniformly selected per row of weld nails, whereas in the door area, eight points per row are designated for measurement. The anchor gauge for prestressed anchor monitoring delivers real-time tension data. Strain monitoring of the external concrete structure entails embedding concrete strain gauges within the cast-in-place concrete. The strain gauge for section C1 monitoring is situated at the interface between the new foundation and rock, whereas the device for section C2 monitoring is positioned at the interface between the new and old foundations. The monitoring of internal forces within the foundation reinforcing bars and external concrete sidewall rebars is executed using rebar force meters affixed to the rebars. The rebar meters are positioned on the reinforcing bars (S1 position) and within the external concrete rebars (S2 position), evenly distributed across the four quadrants of the circumference. The schematic diagram illustrating the layout of monitoring points is presented in Fig 3.

### Data collection

Establishing monitoring points enables real-time assessment of the reinforcement effect on the wind turbine foundation within the wind farm, thereby ensuring the stability of the engineering structure and its safe, continuous operation. This provides essential data support for evaluating long-term benefits. The detailed data collection procedure is as follows: 14,000 monitoring data points were gathered from August 1, 2021, to December 25, 2021, at 15-minute intervals. Fig 4 illustrates a partial data collection scenario.

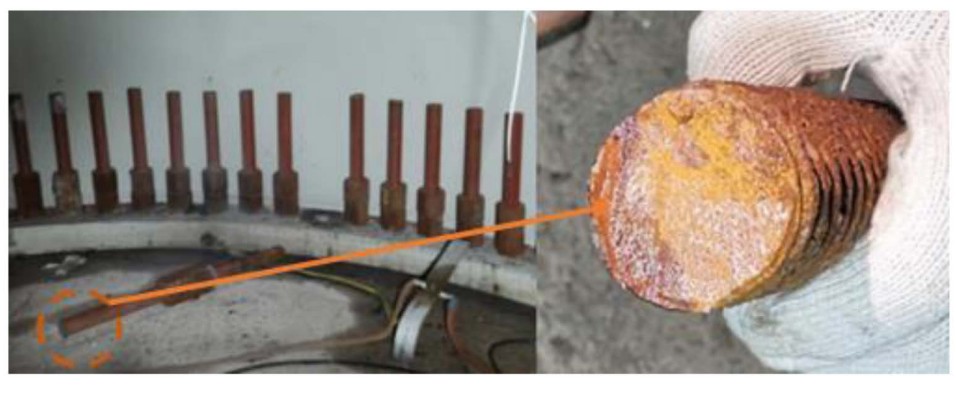

(a)

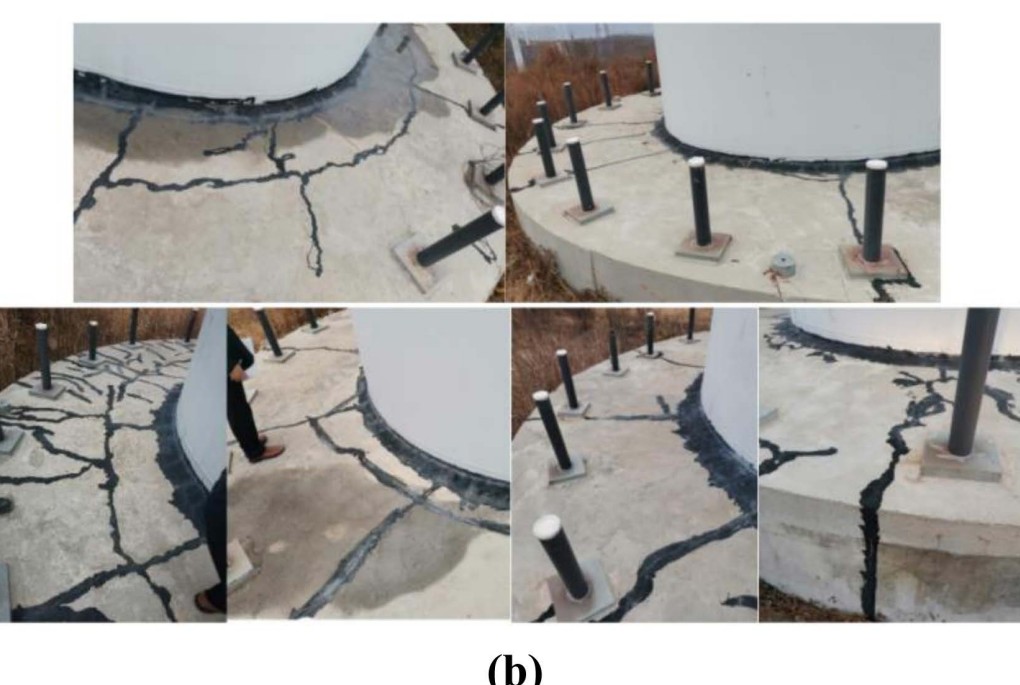

(b)

**Fig 1. Damage to wind turbine foundation, (a) Anchor rod fracture situation, (b) Surface cracks of wind turbine foundation.**

The monitoring data depicted in Fig 4 reveals a significant presence of missing and anomalous values, which impedes data analysis. Therefore, addressing the anomalies and imputing the missing data is essential. To resolve these issues, this paper proposes a method for anomaly detection under missing data conditions and introduces a novel data imputation framework to handle missing values.

## Outlier handling

### Iterative rolling difference-Z-score outlier detection

To tackle the challenge of detecting anomalies in large-scale continuous data caused by sensor failures and massive data loss, this study introduces an innovative iterative rolling difference-Z-score method for anomaly detection. This

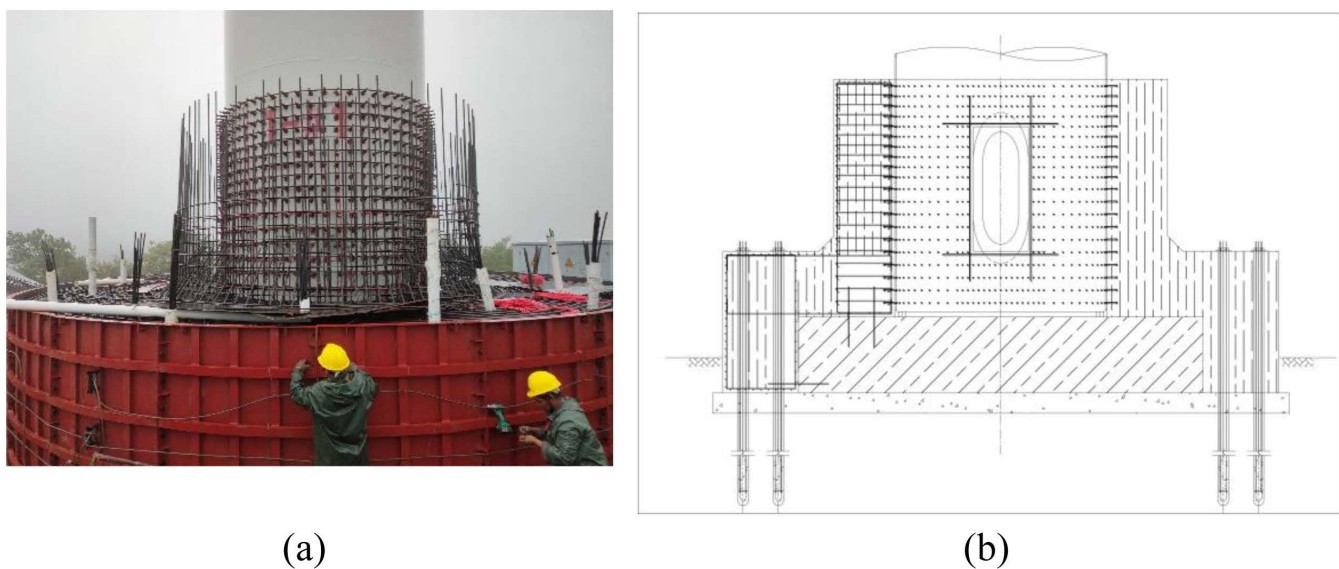

**Fig 2. Schematic diagram of wind turbine reinforcement, (a) Wind turbine reinforcement on-site construction, (b) Wind turbine reinforcement diagram.**

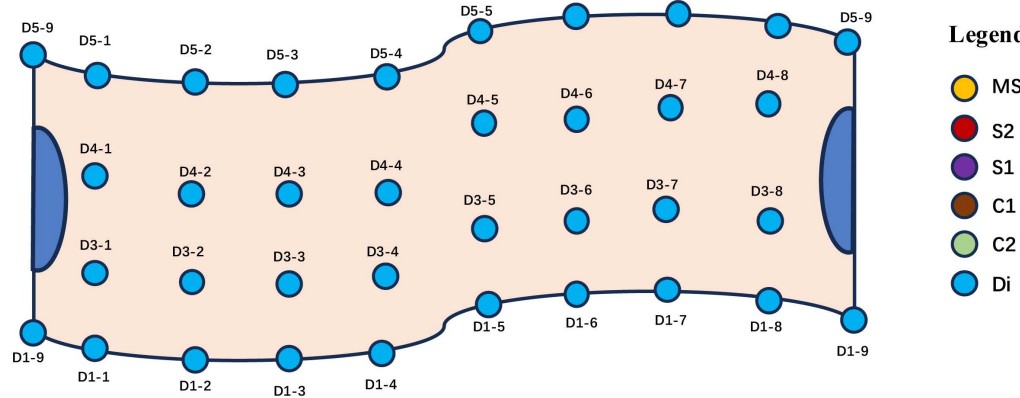

**Wind turbine tower profile**

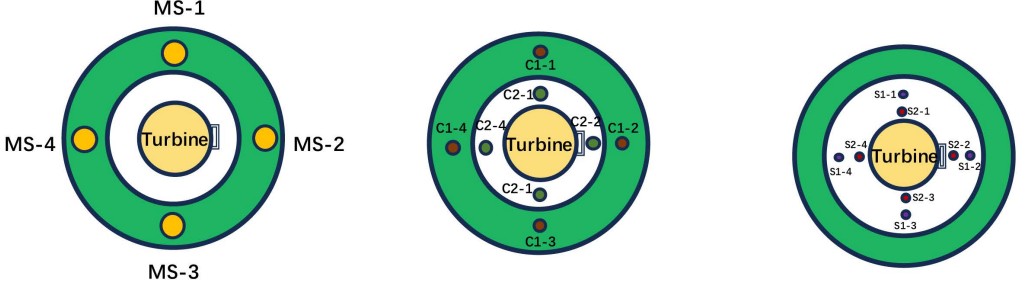

**Top view of fan foundation**

**Fig 3. Layout of monitoring points.**

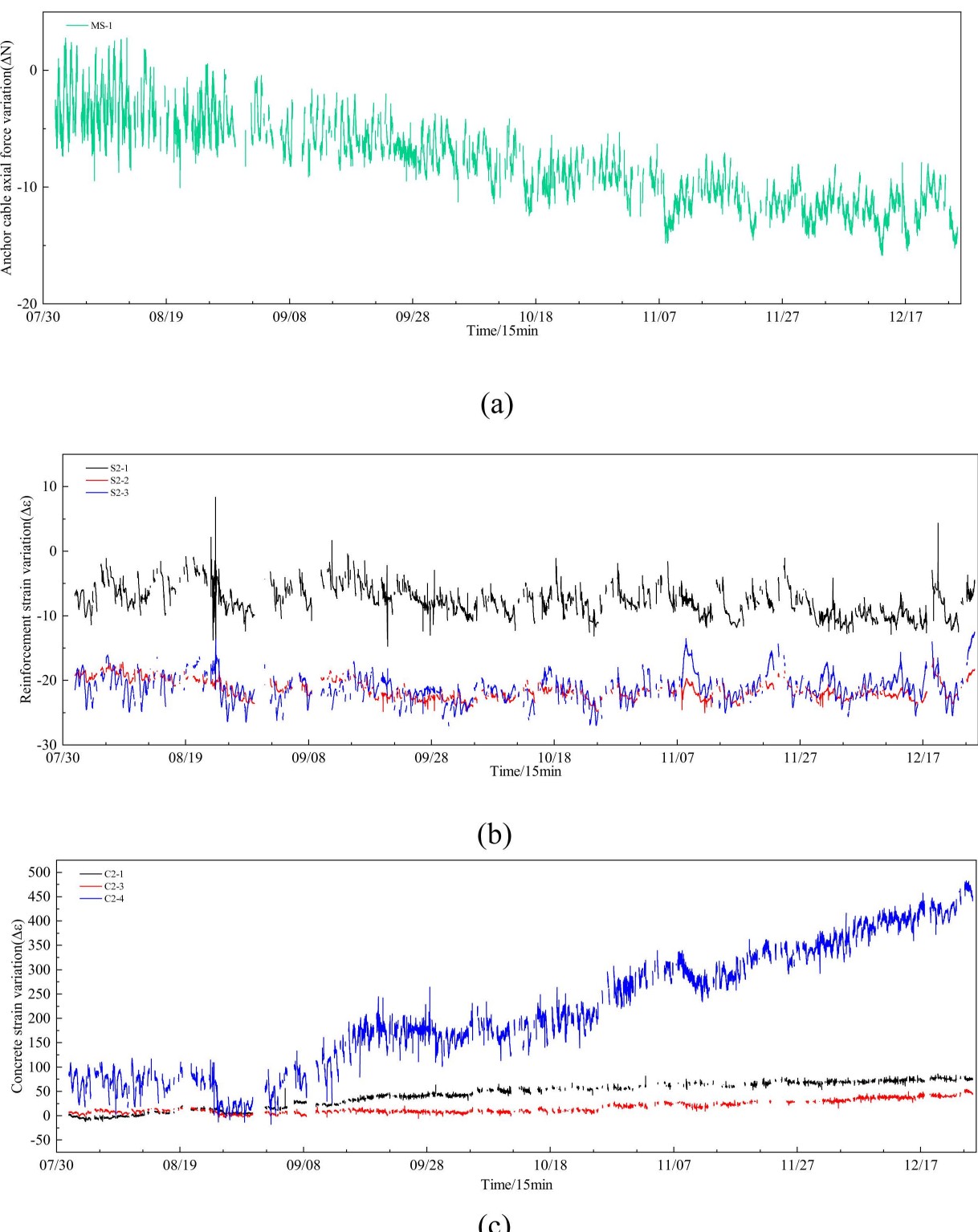

Fig 4. **Partial data variation graph of monitoring, (a) Anchor cable stress variation, (b) Steel bar stress variation, (c) Concrete stress variation.**

method is particularly well-suited for the detection and cleansing of anomalies in time-series data. The methodology proceeds as follows: Initially, rolling differences are computed within a predefined window to effectively accentuate data fluctuations, thereby distinguishing between normal and abnormal variations. Subsequently, Z-scores [10] are employed to transform the difference data, quantifying each data point's deviation from the mean and identifying potential anomalies characterized by significant Z-score values. Following these steps, the rolling difference and Z-score analyses are iteratively performed until anomalies within the dataset are meticulously identified and effectively addressed. This method continuously minimizes the number of anomalies through an iterative optimization process, thereby significantly enhancing overall data quality and analytical accuracy. The anomaly detection process can be represented by the following equation:

$$rolling\_diff\_Z-score = \frac{rolling\_diff - rolling\_diff.mean}{rolling\_diff.std}$$

(1)

In the formula:

*rolling_diff*: Denotes the rolling difference value; *rolling_diff.mean*: Indicates the mean of the rolling difference; *rolling_diff.std*: Signifies the standard deviation of the rolling difference.

We can trace back to the above derivations and see more details about it from Algorithm 1.

```
Algorithm 1. Iterative rolling difference-Z-score outlier detection
Input: DataFrame 'data'
Output: DataFrame 'processed_data'
For each column in 'data':
    1. Drop NaN values from the column, store in 'series'
    2. Initialize 'iterations' to 0
    3. Set 'max_iterations' to 10 # Assuming a maximum of 10 iterations
    4. While 'iterations' < 'max_iterations':
        a. Calculate rolling difference with a specified period, store in 'rolling_diff'
        b. Calculate Z-score for 'rolling_diff'
        c. Set 'threshold' to a predefined value or calculate based on data characteristics
        d. Identify anomalies where abs(Z-score) > 'threshold'
        e. If no anomalies detected, break loop
        f. Filter out anomalies from 'series'
        g. Increment 'iterations' by 1
    5. Add cleaned 'series' to 'processed_data ', reindex with 'data' index
```

## Comparative analysis of anomaly detection models

Presented below is a comparative analysis of anomaly detection capabilities among Iterative Rolling Difference-Z-score, Isolation Forest, One-Class SVM, DBSCAN, LOF (Local Outlier Factor), K-Means, and Gaussian Mixture Model algorithms based on the aggregation feature S1-4 of anomalies (Fig 5).

where (a) – (g) denote Iterative Rolling Difference-Z-score, Isolation Forest, One-Class SVM, DBSCAN, LOF, K-Means, and Gaussian Mixture Model.

As demonstrated in the image above, the Iterative Rolling Difference-Z-score method exhibits superior performance in effectively identifying anomalies compared to other standard anomaly detection algorithms, thereby ensuring the accuracy of monitoring data. Additionally, when compared to manual anomaly removal, the Iterative Rolling Difference-Z-score achieves a correct removal rate of 98.9%, while other algorithms have correct removal rates below 15%. Furthermore, this algorithm offers the advantage of simplified parameter tuning compared to other methods. The anomaly detection parameters for feature S1-4 include three iterations, a rolling difference range of 12, and a Z-score threshold of 4.

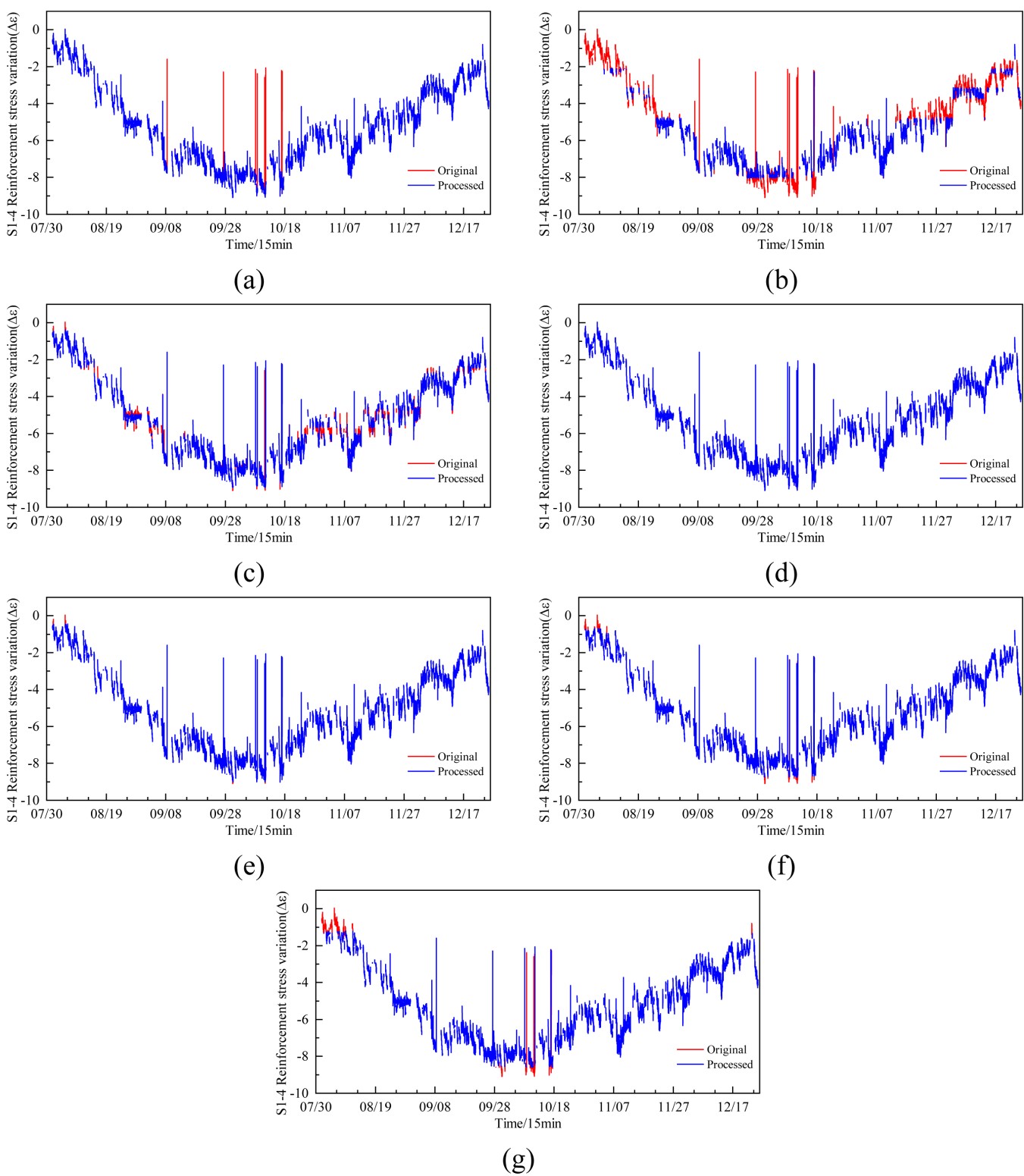

**Fig 5. Illustrates the effectiveness of anomaly detection for features S1-4 across various algorithms, (a) Iterative Rolling Difference-Z-score, (b) Isolation Forest, (c) One-Class SVM, (d) DBSCAN, (e) LOF, (f) K-Means, (g) Gaussian Mixture Model.**

## Experimental analysis of outlier detection models

S1-2 experiences significant challenges with large-scale missing values, including 230 deliberately introduced outlier points that encompass both continuous anomalies and outliers of substantial magnitude. The anomaly removal effectiveness reaches 100% when compared to manual monitoring. The results of the anomaly detection are illustrated in Fig 6.

The anomaly detection parameters for S1-2 data include three iterations, a rolling difference range of 200, and a Z-score threshold of 4. The Iterative Rolling Difference-Z-score method effectively detects clustered anomalies and identifies anomalies in large-scale missing data scenarios. This method is also characterized by its simplicity in adjustment, necessitating only three parameters to achieve optimal results. For datasets with extensive missing values, adjusting the threshold and expanding the rolling difference range can effectively prevent misjudgments of missing data as anomalies, thereby ensuring accurate anomaly detection and maintaining data integrity.

As discussed in the review, the parameter selection for the Iterative Rolling Difference-Z-score method is as follows:

1. Iteration Count Selection:

(1) Frequency of Data Fluctuations: If the data exhibits large fluctuations or contains complex anomaly patterns, increasing the iteration count can help capture these anomalies more effectively. Typically, three iterations are sufficient to smooth the data and extract anomalies, but if there are long-term anomalies or periodic fluctuations within the data, more iterations may be required to improve detection accuracy.

(2) Data Size: For larger datasets, increasing the number of iterations may lead to longer computation times. Therefore, it is essential to balance accuracy with computational efficiency when selecting the iteration count. For smaller datasets, fewer iterations (such as 3) are usually adequate.

(3) Degree of Anomaly Clustering: When a particular segment of the data exhibits significant shifts and a high concentration of anomalies (e.g., 20 or more consecutive anomalies, such as in the S1-4 feature with clustered anomalies), increasing the iteration count is recommended to effectively eliminate these continuous clustered anomalies.

2. Rolling Difference Range Selection:

(1) Data Periodicity or Seasonality: If the data exhibits clear periodic or seasonal patterns, the rolling difference range should match the data's period or seasonal cycle. For data with an annual cycle, a larger rolling difference range

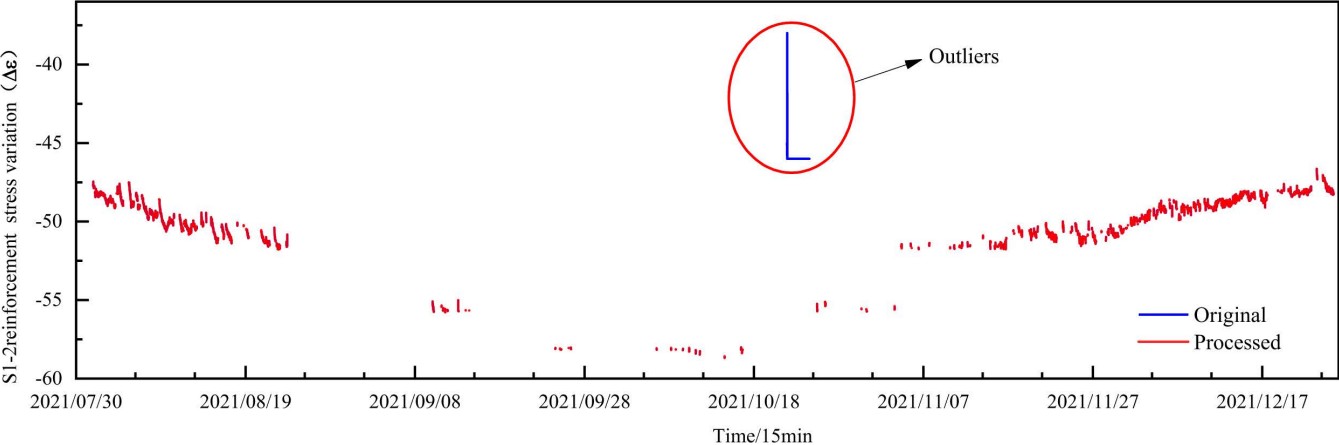

**Fig 6. Detection of large-scale missing values and outliers.**

should be chosen to account for the full seasonal variation. In contrast, for short-term data or data with frequent fluctuations, a smaller rolling difference range (e.g., 12) can improve sensitivity.

(2) Degree of Anomaly Clustering: When anomalies are highly clustered in the data, expanding the rolling difference range can help capture the trend changes of anomalous data more effectively. However, an excessively large difference range might make it difficult to detect smaller anomalies, so adjustments should be made based on the actual data characteristics.

(3) Data Missingness: When a dataset experiences large-scale missing values, it is advisable to increase the rolling difference range to allow the remaining data around the missing values to align better with the overall trend, thereby preventing misclassification of missing data as anomalies.

3. Z-score Threshold Selection:

(1) Data Distribution: The Z-score threshold is primarily used to determine whether a data point is anomalous. A higher Z-score threshold (e.g., 4 or 5) offers a more stringent criterion for anomaly detection, which is suitable for data that is more concentrated. A lower Z-score threshold (e.g., 2 or 3) is more appropriate for data with a more dispersed distribution or for cases where anomalies are less pronounced. Typically, a Z-score threshold around 4 provides a good balance.

(2) Type of Anomalies: If the goal is to strictly detect all types of anomalies, especially those with large magnitudes, lowering the Z-score threshold can improve sensitivity. Conversely, for more subtle or less significant anomalies, raising the Z-score threshold can help reduce false positives.

## Data imputation

Due to the synergistic stresses among various reinforcement components in resisting wind and dynamic loads, inherent relationships and patterns among the monitoring points are evident. Machine learning enables computers to derive patterns and knowledge from data to address various complex problems [30]. Consequently, machine learning methods are employed to drive data-driven approaches for data imputation by discerning the relationships within the data.

The machine learning algorithms employed include Support Vector Machine Regression (SVR), K-Nearest Neighbors Regression (KNN), Ridge Regression, Random Forest Regression, LightGBM, XGBoost, and CatBoost [31–37].

### Data Imputation evaluation indicators

Common evaluation metrics for regression tasks: Mean Squared Error (MSE),Mean Absolute Error (MAE) and Root Mean Squared Error (RMSE)for evaluation:
Mean Squared Error (MSE) calculation formula:

$$MSE = \frac{1}{n} \sum_{i=1}^{n} (y_i - \hat{y}_i)^2$$

(2)

Mean Absolute Error (MAE) calculation formula:

$$MAE = \frac{1}{n} \sum_{i=1}^{n} (|y_i - \hat{y}_i|)$$

(3)

Mean Absolute Error (MAE) calculation formula:

$$RMSE = \sqrt{\frac{1}{n} \sum_{i=1}^{n} (y_i - \hat{y}_i)^2}$$

(4)

Where:

$y_i$ is the actual value data, $\hat{y}_i$ is the imputed value, and $n$ is the number of samples.

## Monitoring data imputation process framework

After analyzing the data, the missing data types are categorized into isolated missing and continuous missing. Fig 7 illustrates the display of the missing data situation.

Corresponding imputation strategies are employed for different types of missing data. Linear interpolation is used for isolated missing data, while machine learning algorithms are utilized for continuous missing data. Based on this, a data imputation framework combining linear interpolation, machine learning, and iterative rolling difference-Z-score algorithm is proposed and divided into two phases.

In the first phase, linear interpolation is used to impute isolated missing data. For columns with relatively few missing values in continuous missing data, machine learning algorithms are used for initial imputation. The imputed data from this phase serves as input for subsequent imputation steps, gradually completing the data for all features through an iterative approach. This phase may introduce potential outliers that need further processing.

In the second phase, a combination of linear interpolation and machine learning methods with the iterative rolling difference-Z-score algorithm is used for outlier detection and imputation. Initially, the iterative rolling difference-Z-score technique is applied to identify and adjust outliers. Subsequently, isolated, missing data is imputed using linear interpolation. Then, a column is chosen as the target variable, and other columns are used as features to predict and fill missing values using machine learning. This process is repeated iteratively until the number of outliers stabilizes. After completing this process, the remaining missing values are finally imputed using the KNN algorithm. Detailed steps of the entire data imputation framework are elaborated in Fig 8 of this paper.

## Monitoring data imputation experiment

To determine the most suitable imputation algorithm, a comparative analysis was performed utilizing Support Vector Machine Regression (SVR), Ridge Regression, Random Forest Regression, LightGBM, XGBoost, and CatBoost. These algorithms were applied to a dataset with missing values omitted, allocating 70% as the training set and 30% as the test set. To ensure reproducibility, the random seed was fixed at 42. Table 1 enumerates the evaluation metrics for the various machine learning algorithms tested, while Table 2 delineates the detailed parameters for these algorithms.

In this experiment, traditional Support Vector Machine (SVM) and Ridge Regression demonstrated poor performance in capturing the complex relationships within the data.However, as illustrated in Table 1, tree-based models excelled in handling nonlinear relationships, with the LightGBM algorithm particularly excelling in the data imputation task.

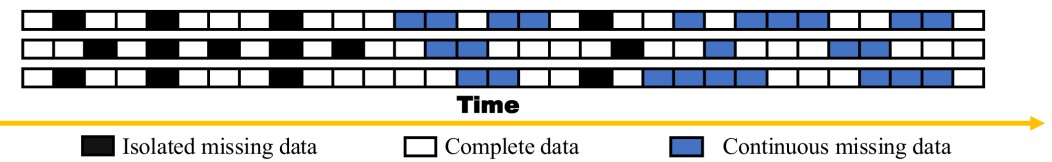

**Fig 7. Visualization of missing data.**

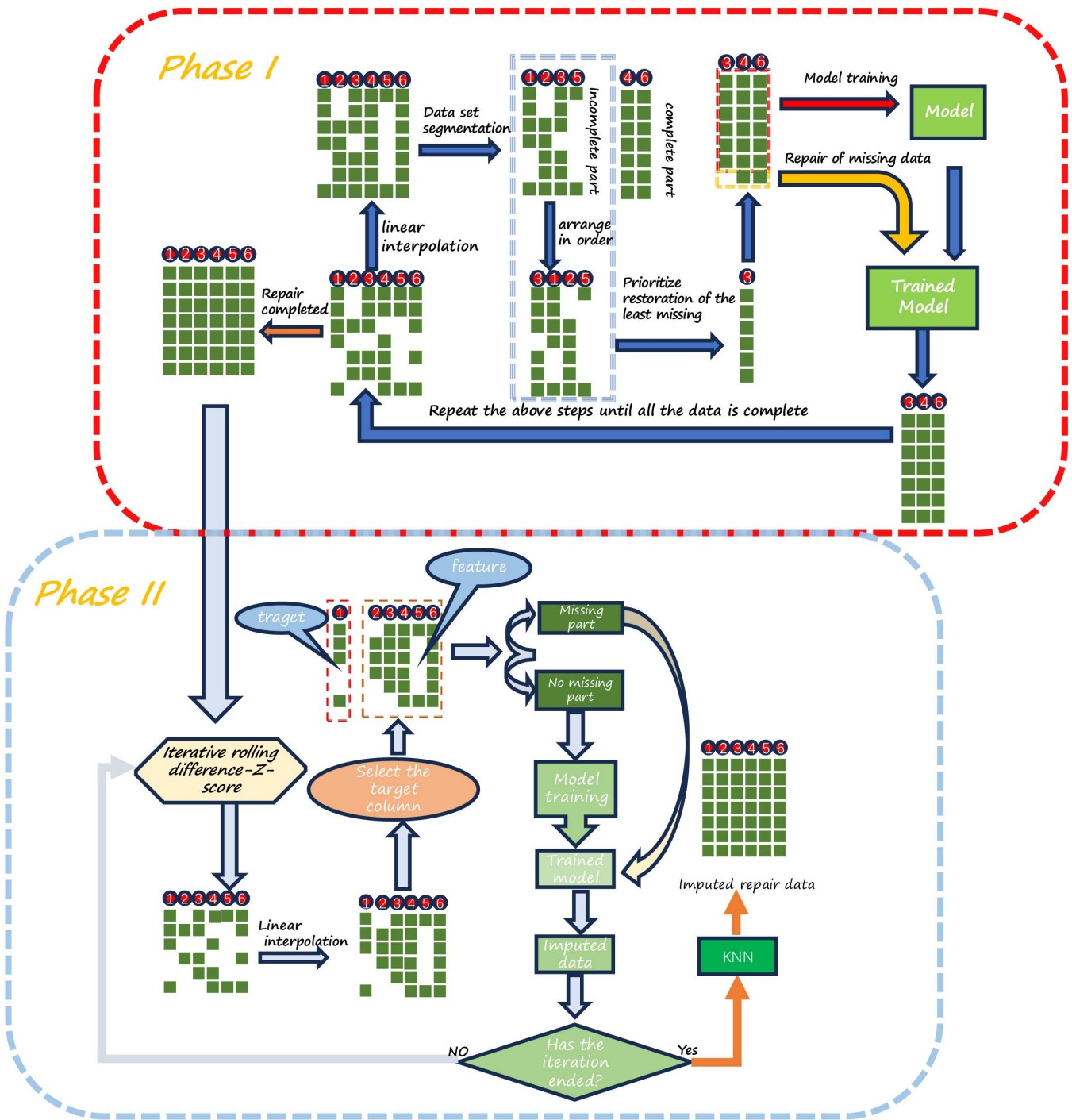

**Fig 8. Data imputation framework.**

To validate the effectiveness of linear interpolation in addressing isolated missing data, the following experiment was conducted: Seven feature columns were randomly selected from the monitoring data, encompassing 1526 data points across 218 rows, with 2% randomly designated as isolated missing values. Linear interpolation was employed to

**Table 1. Performance of different algorithms on the test set.**

| Algorithm | MSE | RMSE | MAE |
|---|---|---|---|
| SVR | 1398.733 | 35.40 | 14.39 |
| Random Forest | 235.71 | 14.36 | 3.45 |
| Ridge Regression | 754.65 | 26.27 | 11.17 |
| CatBoost | 251.82 | 14.65 | 4.53 |
| LightGBM | 228.57 | 15.10 | 3.65 |
| XGboost | 240.22 | 14.90 | 4.00 |

**Table 2. Parameters corresponding to different algorithms.**

| Algorithm | Parameters |
|---|---|
| SVR | Kernel = 'rbf' |
| Random Forest | n_estimators = 1000 |
| Ridge Regression | alpha = 1.0 |
| CatBoost | iterations = 1000, learning_rate = 0.1, early_stopping_rounds = 5 |
| LightGBM | boosting_type = gbdt, n_estimators = 1000 |
| XGBoost | n_estimators = 10000, learning_rate = 0.1, early_stopping_rounds = 5 |

interpolate these isolated missing data points, and the test results before and after imputation were visually compared in Fig 9 The results clearly demonstrate the efficacy of the linear interpolation method in addressing isolated missing data issues and maintaining the accuracy of the original data attributes.

To validate the efficacy of the data imputation framework in addressing continuous missing data, the following experiment was conducted: A feature C1-1, with a missing rate of 32.6% in the original data, was selected as a case study, and the LightGBM algorithm was utilized. For this feature, various degrees of continuous missing data were simulated. Notably, in the monitoring data involved in this study, except for individual instances where the data missing rate reached

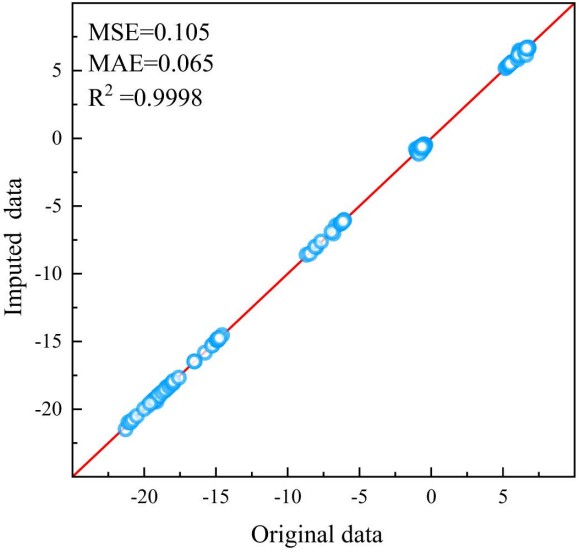

**Fig 9. Comparison chart of linear interpolation repair for individual missing values.**

80%, the continuous missing data did not exceed 180 data points in other cases, averaging approximately 60 continuous missing data points. Therefore, this experiment manually set continuous missing quantities of 60, 120, and 200 on the originally missing data set, with Mean Squared Error (MSE) serving as the evaluation metric. The results are presented in the Table 3 below and Fig 10 clearly indicate that the method proposed in this study exhibits good imputation performance for such missing data, demonstrating the feasibility and effectiveness of the approach.

## Data repair effect and evaluation

In applying the imputation framework, certain noisy data (i.e., outliers) were intentionally retained to bolster the model's robustness. The objective was to develop a model capable of making accurate predictions even when confronted with poor-quality input data. With this aim, the first phase of data imputation was executed.

Due to the high volatility of the data and the retention of original noise during training, there was a potential risk of the algorithm introducing significant bias for certain missing data. Through qualitative analysis of data volatility and the application of the iterative rolling difference-Z-score algorithm, the noise and biased data generated in the first phase were effectively filtered out. The imputation work for the second phase proceeded with the LightGBM algorithm until the number of outliers no longer decreased. In the data imputation using LightGBM, the non-missing data is divided into 80% for the training set, 20% for the test set, and the missing values are used as the validation set for imputation. As the number of iterations increased, the count of outliers gradually decreased, eventually stabilizing (see Fig 11(a)). Upon reaching stability, the remaining small amount of missing data was imputed using the KNN algorithm. The results indicated that the noisy data was effectively removed, leaving only a small number of outliers that were challenging for the second-phase algorithm to learn. Consequently, feature autocorrelation imputation was performed on these outliers using the KNN algorithm. To ascertain the optimal number of neighbors for the KNN application, the study introduced 1% random missing values to the data imputed in the first phase and established a K value search range of [1,20].

Table 3. Evaluation of different missing data scenarios.

| Number of missing values | MSE | RMSE | MAE |
|---|---|---|---|
| 60 | 0.0214 | 0.1400 | 0.121 |
| 120 | 0.0211 | 0.1430 | 0.119 |
| 200 | 0.0227 | 0.1500 | 0.123 |

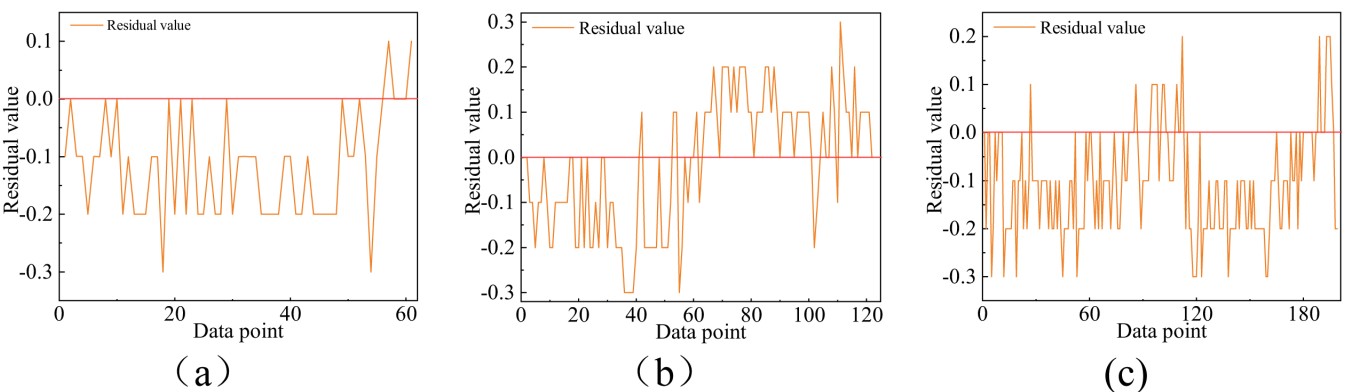

（a）　　　　　　　　（b）　　　　　　　　（c）

Fig 10. Residual plots under different conditions (a) 60 missing data points (b) 120 missing data points (c) 200 missing data points.

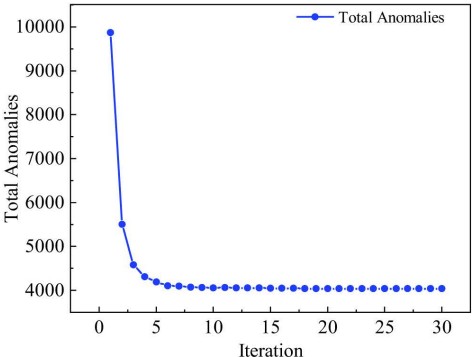 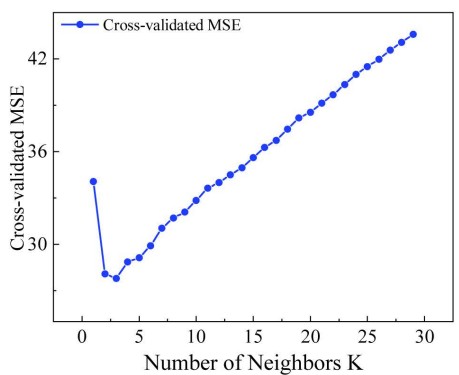

(a) The reduction of outliers with iterations

（b）Relationship between K value and MSE

**Fig 11. Phase 2 model parameter test plot (a) The reduction of outliers with iterations, (b) Relationship between K value and MSE.**

Five-fold cross-validation was conducted, utilizing the average MSE on the test set as the evaluation criterion. Upon verification, the relationship between the K value and MSE was obtained (see Fig 11(b)), and the optimal number of neighbors was determined to be 3.

The following are the data imputation results of this study, highlighting the units with the highest missing rates. A comparison is drawn between the imputed data and the original data for units with missing rates of 51% (C2-1), 80% (S1-2), 38% (D5-2), and 33% (MS-3),as shown in Fig 12.

Through the kernel density [38,39] plot, the distribution of data [40,41] before and after imputation can be visualized, assisting in assessing the imputation effect, as shown in Fig 13.

By comparing the mean and variance, the effectiveness of the imputation can be further quantified. The mean and variance are two important statistical characteristics of data distribution. By comparing the mean and variance of the imputed data and the original data, the degree of deviation in the statistical properties of the imputed data can be assessed, as shown in the Fig 14.

For the MS-3 dataset with 33% data missing, the imputed data closely aligns with the original in terms of kernel density distribution, mean, and variance, showing an ideal imputation effect. Similarly, the D5-2 and C2-1 datasets, with data missing rates of 38% and 51% respectively, exhibit imputed data that generally matches the original in kernel density plots with minor differences in mean and variance, indicating good imputation results. However, for the S1-2 dataset with a high data missing rate of 80%, the imputed data shows noticeable deviations in the kernel density distribution compared to the original data, especially in areas with significant data loss. Although there are overall shortcomings, the imputation results are still relatively ideal in certain parts.

## Model comparison

This paper compares the data imputation framework with common imputation methods, including Multiple Imputation by Chained Equations (MICE), Mean Imputation, and K-Nearest Neighbors Imputation (KNN) (see figure) (Fig 15).

By comparing these different methods, the proposed data repair framework in this paper better reflects the data's changing patterns, demonstrating superior performance in data imputation tasks. In contrast, the proposed framework combines linear interpolation and machine learning methods, especially showing significant advantages in high missing rate scenarios. It not only improves prediction accuracy but also effectively handles the nonlinear relationships and complex patterns in the data.

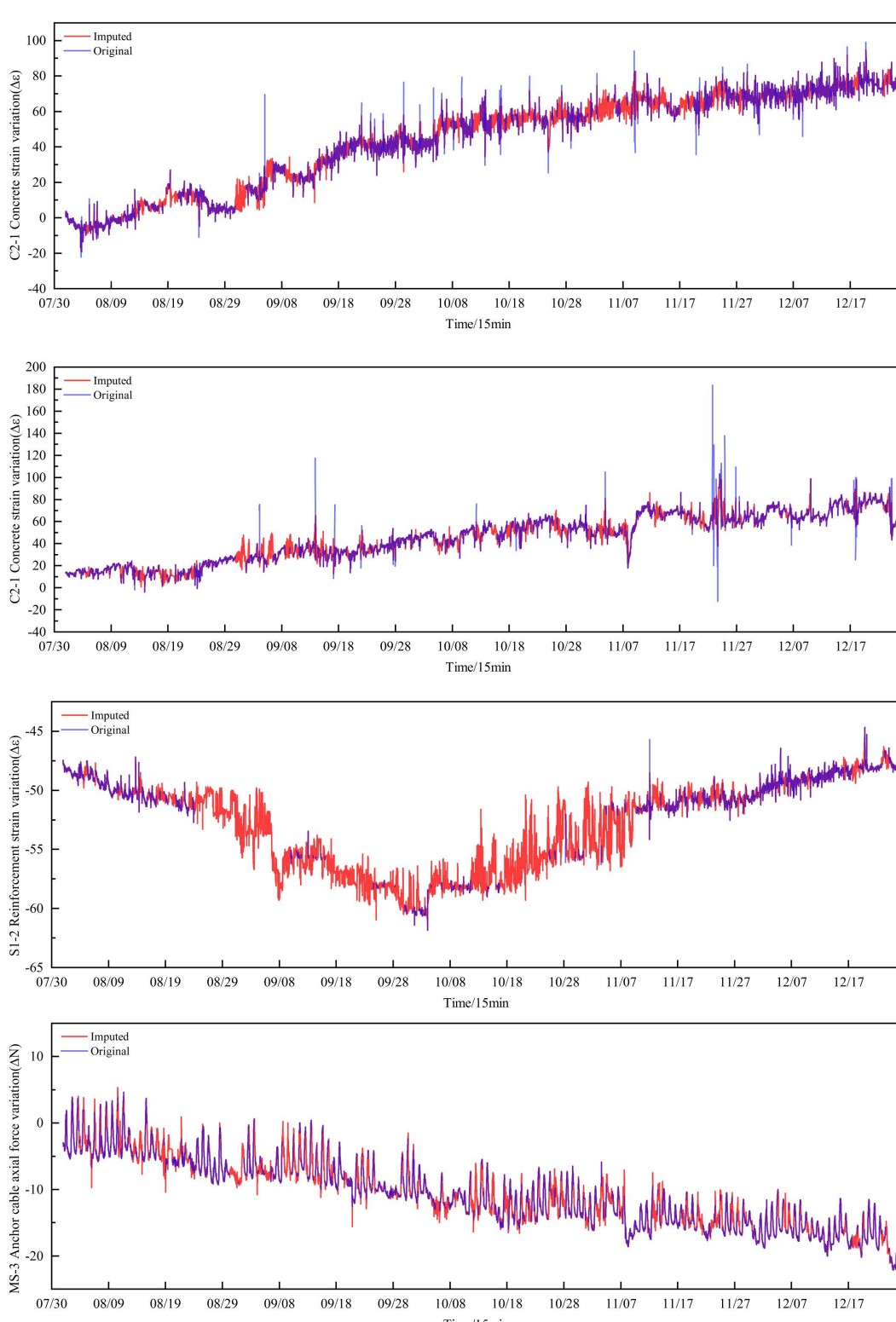

**Fig 12. Comparison of imputed data with original data.**

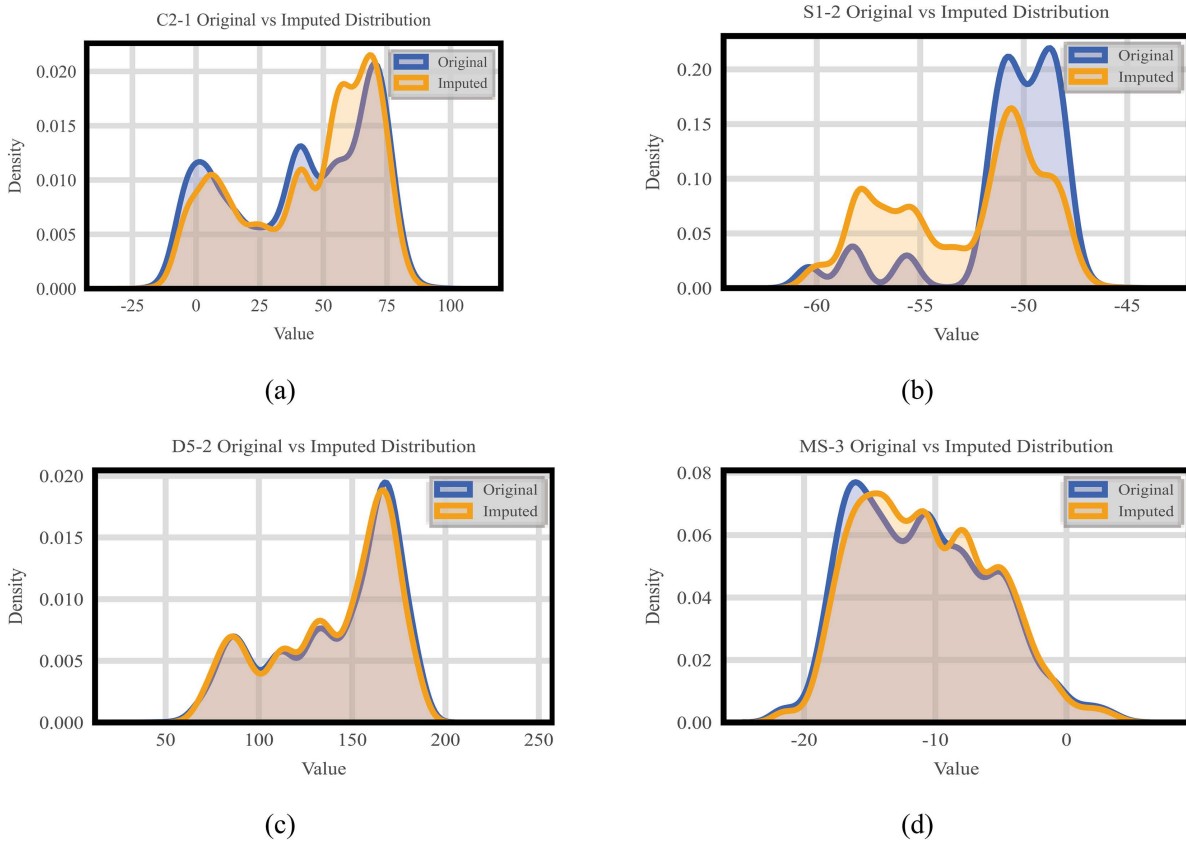

**Fig 13. Kernel density plots of imputed data vs. Original data (a)51% Data Missing (C2-1), (b) 80% Data Missing (S1-2), (c) 38% Data Missing (D5-2), (d) 33% Data Missing (MS-3).**

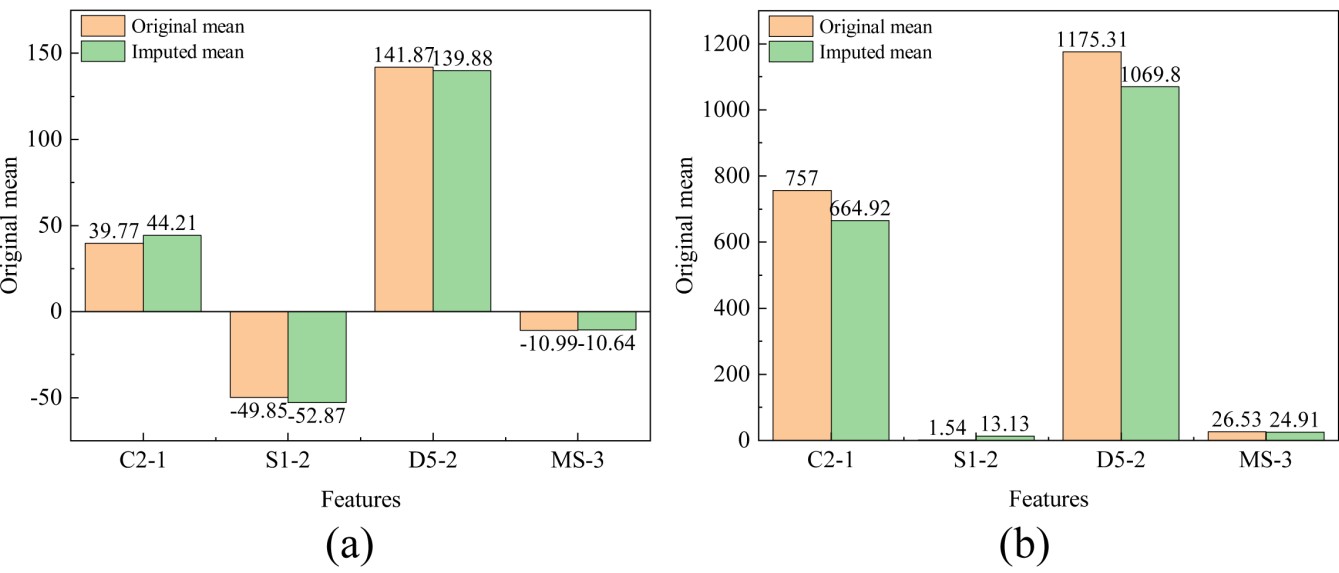

**Fig 14. Comparison of mean and variance between imputed data and original data.**

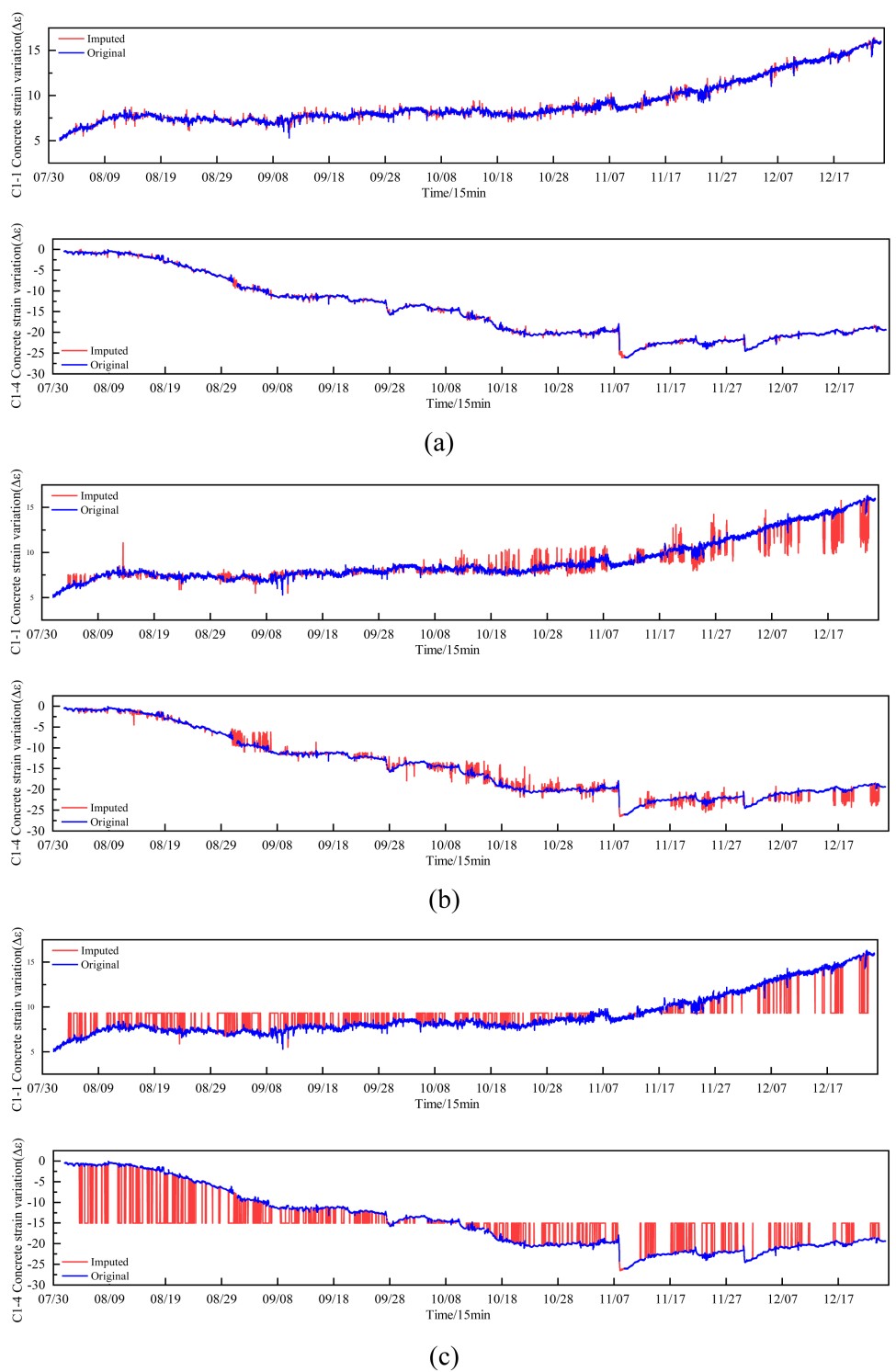

(a)

(b)

(c)

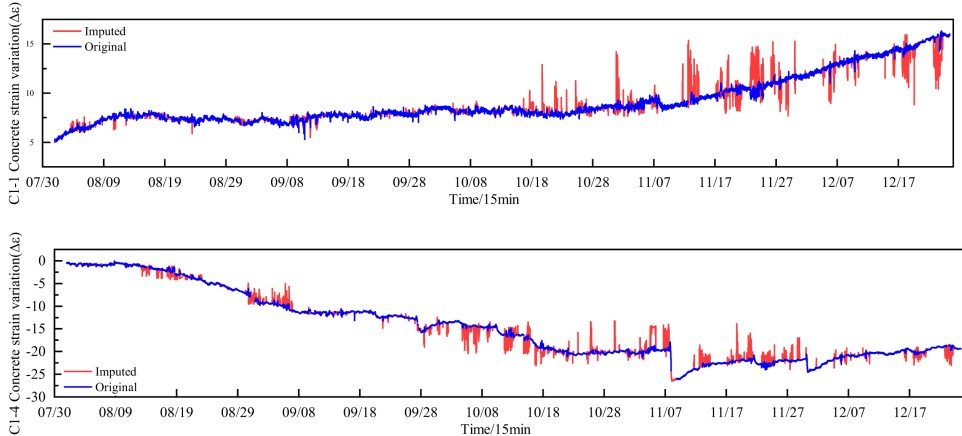

**Fig 15. Comparison of mean and variance between imputed data and original data.**

## Conclusion

This study addresses critical challenges in long-term wind turbine foundation monitoring—data anomalies and missing values—through innovative methodologies that advance the field of structural health monitoring in renewable energy. Within the context of a wind farm reinforcement project in Shandong Province, we introduce a novel iterative rolling difference-Z-score method for anomaly detection and a pioneering data imputation framework integrating linear interpolation with machine learning (LightGBM). These methods offer significant improvements over existing approaches by providing interpretable, transferable, and robust solutions for real-time monitoring data, ensuring the accuracy and reliability essential for assessing turbine foundation health and extending operational longevity.Key findings include:

(1) **Anomaly Detection**: The iterative rolling difference-Z-score method, a novel contribution, excels in detecting single-point and clustered anomalies, maintaining high accuracy even with 80% data loss, unlike traditional methods (e.g., Isolation Forest, One-Class SVM) that struggle with interpretability and transferability.

(2) **Data Imputation**: Our imputation framework, uniquely combining linear interpolation with LightGBM, achieves superior performance with mean squared error (MSE) of 0.0214–0.0227 for continuous missing data (60–200 points) and reliable reconstruction up to 50% data loss, addressing limitations of statistical and less robust machine learning methods.

(3) **Framework Performance**: This dual approach ensures comprehensive data integrity, enabling precise structural assessments critical for preventing wind turbine failures. Its adaptability makes it a scalable solution for other renewable energy and infrastructure monitoring applications.

(4) **Broader Impact**: By enhancing data reliability, our methods support the renewable energy industry's push for sustainable, long-term wind power solutions, reducing maintenance costs and improving operational safety across global wind farms.

(5) **Future Work**: Future research will explore advanced models, such as Transformer networks and generative adversarial networks (GANs), to further improve imputation accuracy for complex data patterns, potentially broadening applications to other critical infrastructure systems.

The proposed methods not only address immediate monitoring challenges but also set a new standard for data-driven structural health assessment, with significant implications for the reliability and sustainability of renewable energy infrastructure worldwide.

## Acknowledgments

We sincerely thank all those who provided help and support during this research. We especially appreciate the valuable assistance from the team at Shandong Jianzhu University and the insightful comments and suggestions from the peer reviewers. Additionally, we would like to express our gratitude to all the team members and collaborators who contributed to this study; their efforts were crucial to the success of the research.

## Author contributions

**Conceptualization:** Renjie Li, JiZhang Zhao, Huanwei Wei.

**Data curation:** JiZhang Zhao, Huanwei Wei.

**Formal analysis:** JiZhang Zhao.

**Funding acquisition:** Renjie Li, Xiangxing Lu, Weibing Chen.

**Investigation:** JiZhang Zhao.

**Project administration:** JiZhang Zhao.

**Resources:** JiZhang Zhao.

**Software:** JiZhang Zhao.

**Supervision:** JiZhang Zhao.

**Validation:** JiZhang Zhao.

**Visualization:** JiZhang Zhao.

**Writing – original draft:** JiZhang Zhao.

**Writing – review & editing:** Huanwei Wei, Cong Liu.

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
