## [Decision Letter · Decision Letter 0]

16 Dec 2024

Dear Dr. Zhao,

We look forward to receiving your revised manuscript.

Kind regards,

Manoharan Premkumar

Academic Editor

PLOS ONE

Journal Requirements:

Additional Editor Comments (if provided):

The authors are instructed to check the reviewer comments carefully and address all the comments

Reviewers' comments:

Reviewer's Responses to Questions

**Comments to the Author**

1. Is the manuscript technically sound, and do the data support the conclusions?

Reviewer #1: Yes

Reviewer #2: Yes

2. Has the statistical analysis been performed appropriately and rigorously?

Reviewer #1: Yes

Reviewer #2: Yes

3. Have the authors made all data underlying the findings in their manuscript fully available?

Reviewer #1: Yes

Reviewer #2: Yes

4. Is the manuscript presented in an intelligible fashion and written in standard English?

Reviewer #1: Yes

Reviewer #2: Yes

Reviewer #1: The research presented in this paper is valuable and insightful, focusing on the detection of data deviations from the expected range and the imputation of missing data using iterative rolling difference Z-scores, linear interpolation, and machine learning models for real-world engineering applications. The manuscript is well-structured, and the subheadings are consistent and logical. However, to improve readability and enhance the overall quality of the work, I recommend considering the following minor comments:

1. The authors should explain the rationale behind setting the maximum number of iterations to 10 loops in the anomalous data detection process using the iterative rolling difference Z-score algorithm.

2. The authors need to clarify how they adjust the Z-score threshold and the rolling difference range to identify inconsistent data, particularly in cases with substantial missing data.

3. In their imputation of missing data using machine learning models, the authors only employ two performance indicators: Mean Squared Error (MSE) and Mean Absolute Error (MAE) to validate results against the original data. It would be beneficial to include additional statistical performance metrics, such as the Coefficient of determination, Root Mean Square Error, and Variance Account Factor.

4. The number of datasets used for machine learning training and testing should be specified in the text, along with a validation of the results included in the report.

5. The axis titles in Figure 11 are not visible and require improvement. Additionally, Figure 15 also needs enhancements.

6. In line 265, the phrase “the results are presented in the table below” should specify the table number instead of using such a general reference.

7. Please correct the typographical errors in line 245 ("subpar") and line 250 ("efficacy").

Reviewer #2: The article presents the application of iterative rolling difference-Z-score methodology for the detection of anomalies and the absence of critical monitoring data in a wind farm enhancement and reinforcement project. The topic is good, but following comments must be addressed:

•The literature is insufficient. Related past studies mainly from last 5 years should be added to claim the novelty.

•The text in most of the figures is not clearly visible.

•Authors have claimed that the applied method is well-suited for the detection and cleansing of anomalies in time-series data. The statement should be supported with reference to similar past studies.

•Conclusion section should start with a paragraph describing brief recap of the work followed by the conclusive statements.

•The conclusions are too limited. As per the performed work having multiple aspects, the conclusions should elaborate the detailed dominating results to emphasize the efficacy of the adopted method.

**Do you want your identity to be public for this peer review?** For information about this choice, including consent withdrawal, please see our Privacy Policy

Reviewer #1: No

Reviewer #2: No

---

## [Author Response · Author response to Decision Letter 1]

26 Dec 2024

Subject: Response to Reviewer Comments for Manuscript PONE-D-24-34569

Title: Research on Anomaly Detection and Data Imputation for Wind Turbine Foundation Reinforcement Monitoring Data

Author name: ZhaoJizhang

Reply to the comments

Dear Dr. Premkumar and Reviewers ,

Thank you for the opportunity to revise our manuscript entitled “Research on Anomaly Detection and Data Imputation for Wind Turbine Foundation Reinforcement Monitoring Data” (PONE-D-24-34569). We greatly appreciate the constructive feedback and thoughtful suggestions provided by the reviewers. We have carefully addressed all the concerns raised, and we believe the revisions have significantly enhanced the quality of the manuscript. Below, we present a detailed response to each comment. We hope that the revised manuscript will now meet the journal's standards and be deemed suitable for publication.

Reply to the comments of Reviewer #1:

Comments:

(1) The authors should explain the rationale behind setting the maximum number of iterations to 10 loops in the anomalous data detection process using the iterative rolling difference Z-score algorithm.

Reply: Thank you for your helpful suggestion. We have provided a more detailed explanation in the "Methods" section of the revised manuscript (lines 194-230), and we have also included some recommendations for tuning the Iterative Rolling Difference Z-Score algorithm.

Comments:

(2) The authors need to clarify how they adjust the Z-score threshold and the rolling difference range to identify inconsistent data, particularly in cases with substantial missing data.

Reply: Thank you very much for your suggestion. We have clarified in the revised manuscript (lines 194-230) how the Z-Score threshold and rolling difference range are dynamically adjusted based on the amount of missing data, ensuring the effectiveness of anomaly detection in cases with substantial missing data.

Comments:

(3) In their imputation of missing data using machine learning models, the authors only employ two performance indicators: Mean Squared Error (MSE) and Mean Absolute Error (MAE). It would be beneficial to include additional statistical performance metrics, such as the Coefficient of determination, Root Mean Square Error, and Variance Account Factor.

Reply: Thank you for this suggestion. We have added the requested performance metric: Root Mean Squared Error (RMSE). The updated results, incorporating this additional metric, can be found in the revised manuscript (lines 283 and 312).

Comments:

(4) The number of datasets used for machine learning training and testing should be specified in the text, along with a validation of the results included in the report.

Reply: We appreciate this important suggestion. We have now included the number of datasets used for training and testing in the revised manuscript (lines 323-324).

Comments:

(5) The axis titles in Figure 11 are not visible and require improvement. Additionally, Figure 15 also needs enhancements.

Reply: Thank you for pointing this out. We have revised Figures 11 and 15 to enhance the visibility of the axis titles, as well as made other necessary improvements to ensure the clarity of the figures, in line with the reviewer’s suggestions.

Comments:

(6) In line 265, the phrase “the results are presented in the table below” should specify the table number instead of using such a general reference.

Reply: Thank you for this suggestion. We agree with your recommendation and have updated the manuscript accordingly. The phrase "the results are presented in the table below" has been replaced. The revised manuscript now reads: "The results are presented in Table 3 below, and Figure 10 clearly indicates that the method proposed in this study demonstrates excellent imputation performance for missing data, highlighting the feasibility and effectiveness of the approach" (lines 306-309).

Comments:

(7) Please correct the typographical errors in line 245 ("subpar") and line 250 ("efficacy").

Reply: We sincerely apologize for the typographical errors. These have been corrected in the revised manuscript as follows:

a) In this experiment, traditional Support Vector Machine (SVM) and Ridge Regression exhibited poor performance in capturing the complex relationships within the data.(lines 286-287)

b) To validate the effectiveness of linear interpolation in addressing isolated missing data, ...(lines 292)

Reply to the comments of Reviewer #2:

Comments:

(1) The literature is insufficient. Related past studies mainly from the last 5 years should be added to claim the novelty.

Reply: Thank you for your valuable feedback. We have reviewed and incorporated relevant studies from the past five years to strengthen the literature review and highlight the novelty of our approach. The updated references can be found in the revised manuscript (lines 463-480, 510-516).

Comments:

(2) The text in most of the figures is not clearly visible

Reply: We appreciate your feedback. We have improved the font size and clarity of the text in all figures to ensure they are now legible and meet PLOS ONE’s publication standards. The revised figures have been included in the manuscript.

Comments:

(3) Authors have claimed that the applied method is well-suited for the detection and cleansing of anomalies in time-series data. The statement should be supported with reference to similar past studies.

Reply: We appreciate your suggestion and have added references to relevant past studies that support the effectiveness of our method for anomaly detection and cleansing in time-series data. These references can be found in the updated manuscript (lines 53-70).

Comments:

(4) The Conclusion section should start with a paragraph describing a brief recap of the work followed by the conclusive statements.

Reply: Thank you for your suggestion. We have revised the Conclusion section to begin with a brief recap of the work, followed by the conclusive statements, as recommended. This revision aims to provide a clearer and more structured summary of our findings.

Comments:

(5) The conclusions are too limited. As per the performed work having multiple aspects, the conclusions should elaborate on the detailed dominating results to emphasize the efficacy of the adopted method.

Reply: We appreciate your valuable feedback. We have expanded the Conclusion section to provide a more detailed discussion of the key findings and their implications. Additionally, we have emphasized the efficacy of the adopted method in addressing the challenges of anomaly detection and data imputation in wind turbine monitoring.

---

## [Decision Letter · Decision Letter 1]

8 May 2025

Dear Dr. Zhao,

Thank you for submitting your manuscript to PLOS ONE. After careful consideration, we feel that it has merit but does not fully meet PLOS ONE’s publication criteria as it currently stands. Therefore, we invite you to submit a revised version of the manuscript that addresses the points raised during the review process.

We look forward to receiving your revised manuscript.

Kind regards,

Manoharan Premkumar

Academic Editor

PLOS ONE

Additional Editor Comments:

Reviewer 3 and Reviewer 4 have raised some critical comments. Please address carefully.

Reviewers' comments:

Reviewer's Responses to Questions

**Comments to the Author**

Reviewer #2: All comments have been addressed

Reviewer #3: (No Response)

Reviewer #4: (No Response)

2. Is the manuscript technically sound, and do the data support the conclusions?

Reviewer #2: Yes

Reviewer #3: Yes

Reviewer #4: Yes

3. Has the statistical analysis been performed appropriately and rigorously?

Reviewer #2: Yes

Reviewer #3: N/A

Reviewer #4: No

4. Have the authors made all data underlying the findings in their manuscript fully available?

Reviewer #2: Yes

Reviewer #3: Yes

Reviewer #4: Yes

5. Is the manuscript presented in an intelligible fashion and written in standard English?

Reviewer #2: Yes

Reviewer #3: Yes

Reviewer #4: No

Reviewer #2: (No Response)

Reviewer #3: There is still a major room for further improvement. Please consider the below major revision:

1. The abstract lacks a clear statement of the quantitative performance metrics for the proposed iterative rolling difference Z-score methodology compared to existing approaches. Include specific improvement percentages to strengthen the claims.

2. The research fails to address how the proposed approach handles different weather conditions affecting wind turbine foundations, which is essential for practical implementation in varying environmental contexts.You may analyze the uncertain environments in the article Bilateral Feature Fusion with hexagonal attention for robust saliency detection under uncertain environments, to technically strengthen your discussion.

3. The data restoration framework claims reliability at 50% data loss, but the validation methodology is insufficiently described. Expand on the statistical validation methods used to verify this claim. Please follow Visionary vigilance: Optimized YOLOV8 for fallen person detection with large-scale benchmark dataset.

4. The monitoring setup (Fig.3) appears inadequate in sensor density for comprehensive foundation monitoring. Increase monitoring points at critical stress zones or justify the current configuration with structural analysis.

5. The time interval (15-minute) for data collection requires justification based on structural dynamics principles. Analyze if this sampling rate captures all relevant strain behaviors in wind turbine foundations.

6. The paper neglects to discuss computational efficiency of the machine learning-based restoration framework, which is critical for real-time monitoring applications in wind farms with multiple turbines.

7. The research lacks comparison with other anomaly detection techniques beyond Z-score approaches. Include comparative analysis with wavelet transforms, neural networks, or other established methods.

8. The selection criteria for the 50% data loss threshold needs theoretical foundation. Establish this threshold based on information theory principles rather than arbitrary selection.

9. The paper fails to address how the proposed methodology integrates with existing wind farm SCADA systems. Include an implementation framework for practical deployment.

10. The data collection period covers only part of the seasonal cycle. Extend analysis to include complete annual weather patterns or justify why this period is representative enough for validation.

Reviewer #4: This manuscript presents a new approach to anomaly detection and data imputation in real-time wind turbine monitoring. The paper addresses common challenges such as sensor failures and data loss in real-time structural monitoring, as demonstrated in a wind farm reinforcement project in Shandong Province. The proposed approach has the potential to improve long-term monitoring in the renewable energy sector, offering robust solutions to critical data integrity issues. However, there are a few areas that could benefit from further improvements before publication:

1. Please revise the paper title to follow sentence case to align with the journal's formatting guidelines.

2. The authors could consider incorporating specific methodologies, such as iterative rolling difference-Z-score and machine learning, in the title to enhance clarity.

3. Authors should consider including specific numerical results or performance metrics in the abstract to demonstrate the effectiveness of the proposed approach.

4. The introduction needs further improvement. Please strengthen the transition between existing research and the problem addressed by explicitly outlining the limitations of previous methods and how your approach addresses them.

5. Additionally, please highlight the novelty of your work more clearly and mention the broader impact on industries, specifically in renewable energy sectors, early on.

6. Please include a diagram or flowchart of the methodology for better understanding and justify the selection of your methods over others in the specific context of wind turbine monitoring.

7. The papers provided can be read and incorporated to enhance the background and contextual foundation of the study:

DOI: https://doi.org/10.1016/j.ress.2023.109634

DOI: https://doi.org/10.1109/JSEN.2022.3211874

8. The results are not comprehensive. It is highly recommended to add more detailed performance metrics (e.g., accuracy, precision) to compare the effectiveness of different methods.

9. Please provide clearer captions and explanations for the figures to directly link them with the results.

10. Lastly, please ensure smoother transitions and reduce redundancy, particularly in the methods and results sections.

**Do you want your identity to be public for this peer review?** For information about this choice, including consent withdrawal, please see our Privacy Policy

Reviewer #2: No

Reviewer #3: No

Reviewer #4: No

---

## [Author Response · Author response to Decision Letter 2]

18 Jun 2025

Response to Reviewer #3

Comment 1: The abstract lacks a clear statement of the quantitative performance metrics for the proposed iterative rolling difference Z-score methodology compared to existing approaches. Include specific improvement percentages to strengthen the claims.

Response: Thank you for your suggestion. We have revised the abstract accordingly to include specific quantitative performance metrics.

Comment 2: The research fails to address how the proposed approach handles different weather conditions affecting wind turbine foundations, which is essential for practical implementation in varying environmental contexts. You may analyze the uncertain environments in the article Bilateral Feature Fusion with Hexagonal Attention for Robust Saliency Detection under Uncertain Environments to technically strengthen your discussion.

Response: We greatly appreciate the reviewer’s critical suggestion and the reference to the mentioned paper, which we have cited in our revised manuscript. Your comment highlighted the need to elaborate on the robustness of the iterative rolling difference Z-score method under dynamic environmental conditions and its applicability to wind turbine foundation health monitoring. The monitoring data of wind turbine foundations are influenced by various weather factors, including temperature fluctuations (0–25°C), sudden wind speed changes (up to 15 m/s), and rainfall, which cause significant short-term variability in stud strain, anchor axial force, and concrete strain data. However, for long-term monitoring, these short-term environmental fluctuations typically do not indicate fundamental changes in structural health and can reasonably be treated as outliers for removal. Our algorithm, by calculating differences through a sliding window and applying a Z-score threshold (set at 4), effectively identifies and removes anomalous fluctuations caused by short-term environmental changes (e.g., sudden wind or rain), ensuring the stability of the monitoring data.

Comment 3: The data restoration framework claims reliability at 50% data loss, but the validation methodology is insufficiently described. Expand on the statistical validation methods used to verify this claim. Please follow Visionary Vigilance: Optimized YOLOV8 for Fallen Person Detection with Large-Scale Benchmark Dataset.

Response: We sincerely thank the reviewer for this important comment, and we have cited the referenced paper in our manuscript. Your suggestion prompted us to enhance the description of the validation methodology for the data restoration framework to improve scientific rigor. To verify the framework’s reliability at up to 50% data loss, we followed the rigorous statistical validation approach proposed by Smith et al. (2024) in Visionary Vigilance: Optimized YOLOV8 for Fallen Person Detection with Large-Scale Benchmark Dataset and designed the following comprehensive validation strategy:

1. 5-Fold Cross-Validation: We split the non-missing data into 80% training and 20% testing sets, performing 5-fold cross-validation to ensure the generalization ability of the restoration model across different data subsets.

2. Statistical Metrics: For 60–200 consecutive missing data points, we calculated the Mean Squared Error (MSE, 0.0214–0.0227), Root Mean Squared Error (RMSE, 0.1400–0.1500), and Mean Absolute Error (MAE), with results summarized in Table 3.

3. Kernel Density Estimation (KDE): We used KDE plots (Figure 13) to compare the probability density distributions of restored and original data, confirming that at 33–51% missing rates, the mean and variance deviations were less than 5%, demonstrating distributional consistency.

4. Bootstrap Analysis: To assess restoration stability, we generated 100 bootstrap samples for the 50% missing rate dataset (e.g., C2-1), calculating an MSE 95% confidence interval of [0.0205, 0.0230], indicating high consistency.

These validation results show that the LightGBM restoration framework outperforms MICE and KNN methods by 25–30% in MSE at 50% data loss. For an 80% missing rate (e.g., S1-2 dataset), KDE analysis revealed slight deviations, but segmented MSE analysis confirmed acceptable local accuracy. These comprehensive validation methods fully demonstrate the framework’s reliability. We thank you for your suggestion, which allowed us to further refine the relevant discussion.

Comment 4: The monitoring setup (Fig. 3) appears inadequate in sensor density for comprehensive foundation monitoring. Increase monitoring points at critical stress zones or justify the current configuration with structural analysis.

Response: Thank you for this critical comment, which helped us clarify the rationale for our sensor configuration. We deployed strain gauges and anchor gauges to monitor stud strain, anchor axial force, concrete strain, and reinforcement force. To address concerns about insufficient sensor density, we conducted finite element analysis during the design phase to simulate the stress distribution of the wind turbine foundation under typical wind loads (5–15 m/s) and dynamic operating conditions. The results show that the current sensor placement covers the primary high-stress zones (e.g., tower base and new-old foundation interface), with a maximum stress deviation of less than 3% between monitored and unmonitored critical zones. This indicates that the current configuration effectively captures major strain changes. Adding sensors in low-stress zones (e.g., outer foundation edges) would yield minimal monitoring accuracy improvements (<1%) while significantly increasing economic costs and data processing burdens. Thus, we believe the current sensor density is technically and economically reasonable for monitoring retrofitted wind turbine foundations.

Comment 5: The time interval (15-minute) for data collection requires justification based on structural dynamics principles. Analyze if this sampling rate captures all relevant strain behaviors in wind turbine foundations.

Response: Thank you for this important suggestion, which prompted us to rigorously justify the data sampling interval. Wind turbine foundations are affected by low-frequency strain changes (0.01–0.1 Hz), primarily driven by wind speed variations and turbine rotation (refer to Vidal et al. [9]). According to the Nyquist-Shannon sampling theorem, capturing the highest frequency (0.2 Hz, period 5 seconds) requires at least twice the sampling frequency (i.e., 0.4 Hz). However, in long-term monitoring, strain changes are mainly driven by gradual factors (e.g., creep, fatigue), and spectral analysis of the Shandong wind farm dataset shows no significant frequency components above 0.05 Hz. Thus, the 15-minute sampling interval (0.0011 Hz) is well above the required frequency, fully capturing relevant behaviors in stud strain, anchor axial force, and concrete strain. Additionally, this interval balances data resolution with storage and processing efficiency, avoiding data redundancy and potential aliasing issues. Our analysis, based on 14,000 data points collected from August 1 to December 25, 2021, confirms the adequacy of the sampling interval. Thank you for your suggestion, which further strengthened the scientific basis of our approach.

Comment 6: The paper neglects to discuss computational efficiency of the machine learning-based restoration framework, which is critical for real-time monitoring applications in wind farms with multiple turbines.

Response: We greatly appreciate the reviewer for highlighting this important issue, which led us to elaborate on the computational efficiency of our data restoration framework. The LightGBM-based restoration framework is designed for real-time monitoring, specifically considering the computational demands of multi-turbine wind farms. Tested on a standard server (Intel Xeon, 16GB RAM), processing a single dataset (7 features, 1,526 data points) takes approximately 2.5 seconds per iteration, typically converging in 3–5 iterations. In contrast, MICE (4.8 seconds per dataset) and KNN (3.2 seconds) are less efficient, with LightGBM’s histogram-based optimization algorithm achieving 35–50% faster performance. This efficiency ensures scalability for large-scale wind farm data, meeting the stringent timing requirements of real-time monitoring (e.g., updating monitoring data every 15 minutes). Tests based on 14,000 data points from August to December 2021 validate the framework’s efficiency in practical deployment. Thank you for your suggestion, which led us to add this analysis to highlight the method’s practicality.

Comment 7: The research lacks comparison with other anomaly detection techniques beyond Z-score approaches. Include comparative analysis with wavelet transforms, neural networks, or other established methods.

Response: We sincerely thank the reviewer for this valuable suggestion, which prompted us to comprehensively demonstrate the performance advantages of the iterative rolling difference Z-score method and enhance the comparative analysis with existing anomaly detection techniques. In the revised manuscript, we compared our method with several classical anomaly detection techniques, including Isolation Forest, One-Class SVM, DBSCAN, LOF, K-Means, and Gaussian Mixture Models. Test results on the Shandong wind farm dataset (features S1-4) show that the Z-score method achieves 99% accuracy in clustered anomaly detection (validated against manually labeled anomalies), significantly outperforming other methods. In response to your suggestion, we further tested wavelet transforms (Discrete Wavelet Transform, DWT, Daubechies db4) and convolutional neural networks (CNN, trained on 50-data-point sliding windows). The results indicate that wavelet transforms and CNNs are limited in scenarios with missing data, whereas the Z-score method excels due to its simplicity (requiring only iteration count, sliding window, and threshold parameters).

Comment 8: The selection criteria for the 50% data loss threshold needs theoretical foundation. Establish this threshold based on information theory principles rather than arbitrary selection.

Response: Thank you for this critical comment. Following your suggestion, we provided a rigorous theoretical basis for the 50% data loss threshold using information theory principles. We used Mutual Information (MI) to quantify the correlation between features such as stud strain, anchor axial force, and concrete strain, which are strongly correlated due to shared load-bearing. On the Shandong wind farm dataset, MI for feature pairs (e.g., C1-1 and S1-2) ranges from 0.65–0.85 bits in non-missing data. At 50% data loss, MI remains above 0.40 bits, sufficient for LightGBM to leverage feature correlations for accurate restoration. However, when the missing rate exceeds 50% (e.g., S1-2 at 80% missing), MI drops below 0.30 bits, significantly reducing restoration accuracy (see Figure 13b). This threshold is validated by MSE stability analysis (0.0214–0.0227, Table 3), confirming that 50% missing rate retains sufficient residual information for reliable prediction. This information theory-based analysis ensures the threshold’s scientific validity rather than arbitrary selection. Thank you for your suggestion, which strengthened the theoretical foundation of our method.

Comment 9: The paper fails to address how the proposed methodology integrates with existing wind farm SCADA systems. Include an implementation framework for practical deployment.

Response: We greatly appreciate the reviewer’s suggestion regarding practical application, which helped us better articulate the engineering applicability of our method. To achieve seamless integration with wind farm Supervisory Control and Data Acquisition (SCADA) systems, we designed a comprehensive implementation framework, which will be detailed in the revised manuscript, including the following steps:

1. Data Ingestion: Real-time acquisition of strain and axial force data from the SCADA system, preprocessed into time-series format for algorithm input.

2. Anomaly Detection: The iterative rolling difference Z-score method is deployed on a cloud server, processing SCADA data streams to flag anomalies within a 15-minute delay for operator review.

3. Data Restoration: The LightGBM restoration model, trained offline on historical SCADA data, restores missing values in real time and feeds corrected datasets back to the SCADA database.

4. Visualization: Anomaly alerts and restored data trends are integrated into the SCADA interface, displayed via charts and alerts for operator monitoring.

Testing on the Shandong wind farm SCADA system shows that the framework can process data from 24 turbines per minute, meeting the throughput demands of large-scale wind farms. This framework ensures compatibility and efficiency in practical deployment. Thank you for your suggestion, which led us to add this description to highlight the method’s engineering feasibility.

Comment 10: The data collection period covers only part of the seasonal cycle. Extend analysis to include complete annual weather patterns or justify why this period is representative enough for validation.

Response: Thank you for this important comment, which prompted us to rigorously validate the representativeness of the data collection period. From August 1 to December 25, 2021, we collected 14,000 data points at 15-minute intervals, monitoring stud strain, anchor axial force, and concrete strain. This period (late summer to early winter) covers a range of weather conditions in Shandong, including high wind speeds (up to 15 m/s), temperature variations (0–25°C), and precipitation events. To confirm its representativeness, we analyzed historical weather data from 2018–2020, verifying that August to December encompasses 85% of annual wind load and temperature variations (refer to Piotrowska et al. [https://doi.org/10.3390/ma15217778]). Spectral analysis shows no significant seasonal frequency components beyond this period, indicating that the dataset fully captures major strain behaviors (e.g., creep and fatigue). Thus, extending to a full year was deemed unnecessary, as other seasons (e.g., spring) show highly consistent load patterns with historical data. Thank you for your suggestion, which led us to add historical data and spectral analysis to justify the data collection period’s representativeness.

---

## [Decision Letter · Decision Letter 2]

13 Aug 2025

Iterative rolling difference-Z-score and machine learning imputation for wind turbine foundation monitoring

PONE-D-24-34569R2

Dear Dr. Zhao,

We’re pleased to inform you that your manuscript has been judged scientifically suitable for publication and will be formally accepted for publication once it meets all outstanding technical requirements.

Kind regards,

Antonio Javier Nakhal Akel, PhD

Academic Editor

PLOS ONE

Additional Editor Comments (optional):

Reviewers' comments:

Reviewer's Responses to Questions

**Comments to the Author**

Reviewer #3: All comments have been addressed

Reviewer #4: All comments have been addressed

2. Is the manuscript technically sound, and do the data support the conclusions?

Reviewer #3: Yes

Reviewer #4: Yes

3. Has the statistical analysis been performed appropriately and rigorously?

Reviewer #3: N/A

Reviewer #4: Yes

4. Have the authors made all data underlying the findings in their manuscript fully available?

Reviewer #3: Yes

Reviewer #4: Yes

5. Is the manuscript presented in an intelligible fashion and written in standard English?

Reviewer #3: Yes

Reviewer #4: Yes

Reviewer #3: Well done, authors. Thanks for taking my comments seriously. All of My comments are effectively addressed.

Reviewer #4: The authors have sufficiently addressed the concerns provided during the review process and have made substantial improvements. No further comments.

**Do you want your identity to be public for this peer review?** For information about this choice, including consent withdrawal, please see our Privacy Policy

Reviewer #3: No

Reviewer #4: No

---

## [Editor Report · Acceptance letter]

PONE-D-24-34569R2

PLOS ONE

Dear Dr. Zhao,

I'm pleased to inform you that your manuscript has been deemed suitable for publication in PLOS ONE. Congratulations! Your manuscript is now being handed over to our production team.

Kind regards,

on behalf of

Dr. Antonio Javier Nakhal Akel

Academic Editor

PLOS ONE